# Serotonin enhances excitability and gamma frequency temporal integration in mouse prefrontal fast-spiking interneurons

Jegath C Athilingam[1,2,3,4,5], Roy Ben-Shalom[2,3,4], Caroline M Keeshen[2,3,4], Vikaas S Sohal[1,3,4]*, Kevin J Bender[2,3,4]*

[1]Department of Psychiatry, University of California, San Francisco, San Francisco, United States; [2]Department of Neurology, University of California, San Francisco, San Francisco, United States; [3]Weill Institute for Neurosciences, University of California, San Francisco, San Francisco, United States; [4]Kavli Institute for Fundamental Neuroscience, University of California, San Francisco, San Francisco, United States; [5]Neuroscience Graduate Program, University of California, San Francisco, San Francisco, United States

*For correspondence:
vikaas.sohal@ucsf.edu (VSS);
kevin.bender@ucsf.edu (KJB)

**Competing interests:** The authors declare that no competing interests exist.

**Abstract** The medial prefrontal cortex plays a key role in higher order cognitive functions like decision making and social cognition. These complex behaviors emerge from the coordinated firing of prefrontal neurons. Fast-spiking interneurons (FSIs) control the timing of excitatory neuron firing via somatic inhibition and generate gamma (30–100 Hz) oscillations. Therefore, factors that regulate how FSIs respond to gamma-frequency input could affect both prefrontal circuit activity and behavior. Here, we show that serotonin (5HT), which is known to regulate gamma power, acts via 5HT2A receptors to suppress an inward-rectifying potassium conductance in FSIs. This leads to depolarization, increased input resistance, enhanced spiking, and slowed decay of excitatory post-synaptic potentials (EPSPs). Notably, we found that slowed EPSP decay preferentially enhanced temporal summation and firing elicited by gamma frequency inputs. These findings show how changes in passive membrane properties can affect not only neuronal excitability but also the temporal filtering of synaptic inputs.
DOI: https://doi.org/10.7554/eLife.31991.001

## Introduction

The prefrontal cortex (PFC) organizes higher order cognitive functions ranging from decision making to social cognition (*Euston et al., 2012*; *Dias et al., 1996*; *Ray and Zald, 2012*).These complex behaviors emerge from the coordinated firing of PFC neurons, resulting in neuronal oscillations (*Buzsáki and Chrobak, 1995*; *Buzsáki and Wang, 2012*). Synchronized oscillations of neuronal activity in the gamma frequency range (30–100 Hz) play a key role in information encoding (*Buzsáki and Chrobak, 1995*; *Buzsáki and Wang, 2012*) and prefrontal gamma oscillations influence the performance of tasks related to cognitive flexibility and attention (*Cho et al., 2015*; *Kim et al., 2016*). The neuromodulator serotonin (5HT) has been shown to regulate gamma power in motor cortex (*Puig et al., 2010*), suggesting that it could play a role in regulating task-dependent changes in gamma oscillations. Furthermore, neuropsychiatric disease associated with deficits in PFC gamma synchrony, including schizophrenia and depression, are currently treated with medications that have high affinity for serotonin receptors (*Meltzer and Massey, 2011*). Overall, this suggests that

serotonergic modulation of gamma oscillations is important for prefrontal function; however, the cellular mechanisms by which 5HT modulates gamma oscillations remain elusive.

Gamma oscillations are orchestrated by cortical fast-spiking interneurons (FSIs; *Cardin et al., 2009*; *Sohal et al., 2009*; *Bartos et al., 2007*; *Bartos et al., 2002*; *Galarreta and Hestrin, 2001*). In contrast to neighboring excitatory cells, FSIs resonate intrinsically in the 30–50 Hz range (*Bracci et al., 2003*; *Fellous et al., 2001*; *Pike et al., 2000*). In turn, FSIs are more likely to generate action potentials in response to gamma-modulated sinusoidal waveforms (*Pike et al., 2000*), suggesting that action potential generation in FSIs may favor gamma frequency input. This process may be regulated by 5HT. Indeed, 5HT can increase FSI intrinsic excitability, as measured at the soma (*Weber and Andrade, 2010*; *Zhou and Hablitz, 1999*; *Zhong and Yan, 2011*), but whether this also changes how FSIs encode synaptic input is not known.

Here, we used patch clamp electrophysiology, glutamate uncaging, optogenetic stimulation, and compartmental modeling to investigate how serotonergic modulation of FSIs can regulate the integration of synaptic inputs, particularly at gamma frequencies. We found that 5HT enhances the excitability of FSIs due to a depolarization caused by a suppression of inward-rectifying potassium channels. Furthermore, we found that this reduction of potassium conductance in FSI dendrites increases the time constant of synaptic potentials, leading to a selective enhancement of temporal summation of gamma frequency inputs. This made FSIs more likely to fire action potentials, specifically in response to gamma-frequency inputs, and resulted in more inhibition in the gamma frequency band in downstream pyramidal neurons. These results suggest that 5HT can play a role in modulating prefrontal circuit activity by enhancing the flow of gamma-frequency information through FSIs via modulation of passive membrane properties.

## Results

### Activating 5HT2A receptors increases FSI intrinsic excitability

To determine if 5HT modulated the intrinsic properties of FSIs, we performed whole-cell patch clamp recordings of fluorescent neurons in PV-Cre::Ai14 mice and applied 5HT (30 µM). 5HT consistently depolarized FSIs by $6.1 \pm 1.1$ mV, from $-71 \pm 1.3$ mV to $-66 \pm 2.2$ mV (*Figure 1—figure supplement 1B*, p<0.005, paired t-test 5HT vs. baseline, n = 10; *Figure 1C*, p=0.03, 5HT n = 10 vs. time-locked controls n = 8, post-hoc Tukey comparison after one-way ANOVA with p=*0.005*) and increased input resistance by $31.1 \pm 6.0\%$, from $92.9 \pm 11.5$ MΩ to $121.6 \pm 16.9$ MΩ (*Figure 1—figure supplement 1C*, p=0.026, paired t-test 5HT vs. baseline, n = 10; *Figure 1D*, p=*0.037* 5HT n = 10 vs. time-locked controls n = 8, post-hoc Tukey comparison after one-way ANOVA with p=*0.004*). We found this concentration of 30 µM to be sub-saturating, eliciting approximately 80% of the maximal response (*Figure 1—figure supplement 2*). 5HT depolarized FSIs even in the presence of ionotropic glutamatergic (10 µM CNQX, 100 µM DL-AP5) and GABAergic (10 µM SR95531) antagonists (*Figure 1C–D*, p=*0.043* for $V_m$ and p=*0.05* for $R_{in}$, 5HT + syn block n = 9 vs. time-locked controls n = 8, post-hoc Tukey comparison after one-way ANOVAs with p=*0.005* for $V_{rest}$ and p=*0.004* for $R_{in}$), but effects were blocked (n = 7) by the 5HT2A antagonist MDL100907 (1 µM, *Figure 1C–D*, p=*0.024* for $V_m$ and p=*0.026* for $R_{in}$, 5HT vs. 5HT + 2A antagonist post-hoc Tukey comparison after one- way ANOVAs with p=*0.005* for $V_{rest}$ and p=*0.004* for $R_{in}$). Thus, 5HT signals through 2A receptors expressed on FSIs.

Changes in $V_m$ and $R_{in}$ can change neuronal firing properties. Indeed, 5HT increased spiking in response to somatic current injection (*Figure 1B*; *Figure 1—figure supplement 1D*, p=0.028 for treatment factor, 5HT vs. baseline in repeated measures two-way ANOVA for firing rate vs. current curve with current and treatment as factors, n = 10; *Figure 1—figure supplement 1F*, p = 0.001, rheobase, 5HT vs. baseline, paired t-test, n = 10). No other changes in intrinsic firing properties were noted after 5HT application (*Figure 1—figure supplement 1G–L*).

Two broad classes of cortical interneurons arise from the medial (MGE) and caudal (CGE) ganglionic eminences (*Rudy et al., 2011*). Interneuron classes that arise from the CGE, including CCK and VIP interneurons, have been shown to express 5HT3A receptors exclusively (*Lee et al., 2010*); however, It is unclear whether the other major MGE-derived class, somatostatin-expressing interneurons (SOM), express 5HT2A receptors. Therefore, we made similar excitability measurements from fluorescently identified SOM + neurons (*Figure 1—figure supplement 3*). Intrinsic excitability was

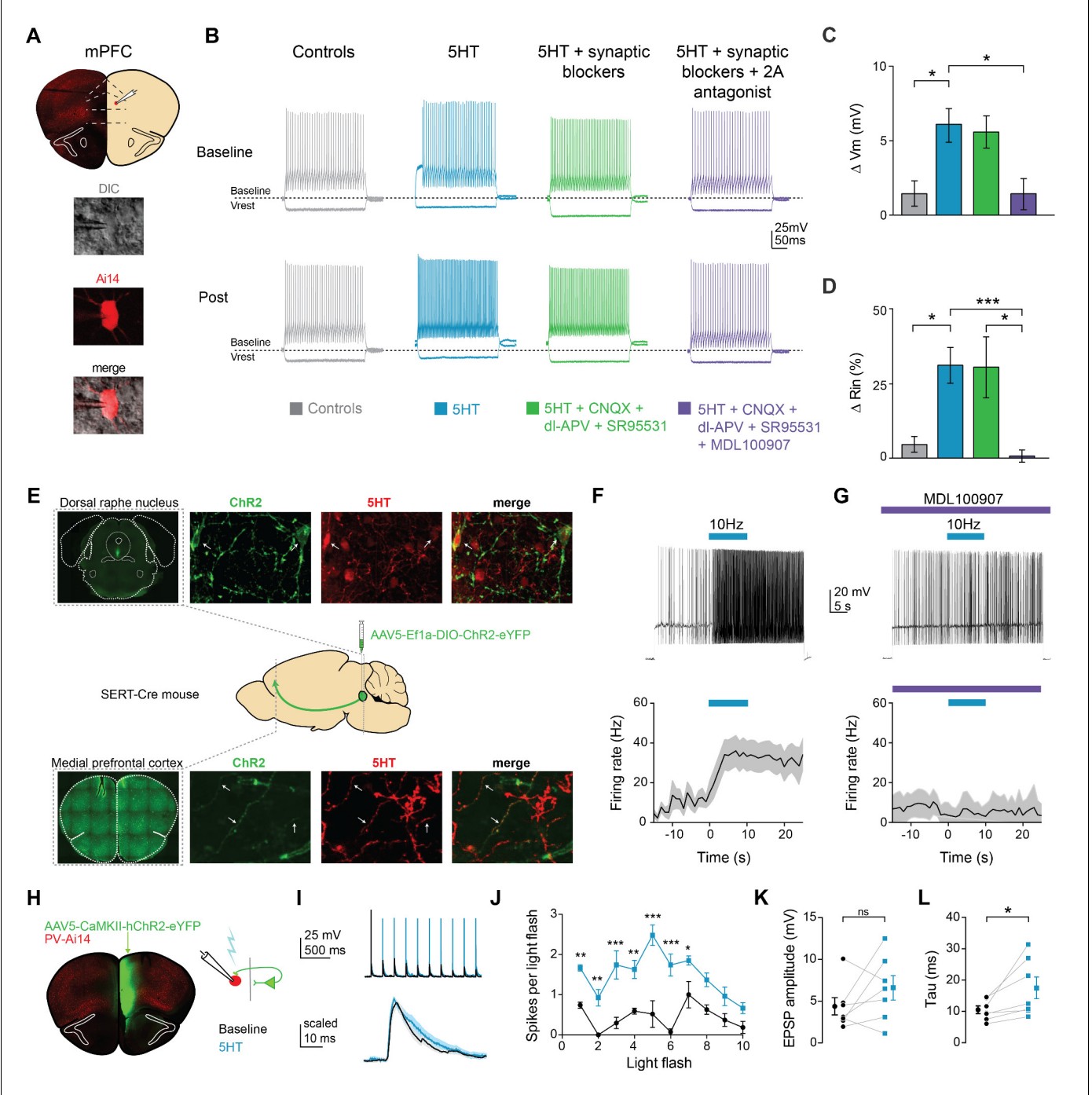

**Figure 1.** Serotonin alters intrinsic properties to increase FSI excitability. (**A**) Experimental design: we recorded from fast-spiking interneurons labeled in a PV-Cre:: Ai14 in mPFC (top). Images of a recorded neuron in DIC and showing tdTomato expression (bottom). (**B**) Example FSI responses to hyperpolarizing (−200 pA) and depolarizing (50 pA above rheobase) current steps at baseline and after application (Post, 10 min after drug wash in) of 5HT (30 µM, blue), 5HT + synaptic blockers (10 uM CNQX, 100 uM DL-AP5, 10 uM gabazine, green), 5HT + synaptic blockers+5HT2A antagonist (1 uM MDL-100907, purple), or time-locked control aCSF (gray). (**C–D**) Subtracted change in membrane potential (**C**) and percent change in input resistance (**D**) after pharmacological manipulations listed above. (**E**) Experimental design: Cre-dependent ChR2 was injected into the dorsal raphe of SERT-Cre mice. Top and bottom rows: Images of ChR2 expression and 5HT immunohistochemistry in dorsal raphe injection side (top) and mPFC recording site (bottom). Confocal images of ChR2 (green), 5HT immunohistochemistry (red), and merged. Yellow sections indicate overlap. Arrows point to examples of overlap. (**F**) Top: FSIs in mPFC were injected with light depolarizing current to elicit spiking and ChR2 expressing terminals were activated with blue light (10 Hz, 10 s) to release endogenous 5HT (top). Bottom: Peristimulus time histograms of FSI firing rate during current step with ChR2-activated release of 5HT (**G**) These experiments were repeated after washing in a 5HT2A antagonist (1 uM MDL100907). (**H**) Experimental design: ChR2 was

*Figure 1 continued on next page*

Figure 1 continued

injected into one hemisphere of mPFC and FSIs were patched on the opposite hemisphere. (I) Example traces of FSI responses at baseline (black) and after 5HT (blue) in response to activation of synaptic inputs from ChR2-expressing terminals with either a train of blue light pulses (5 Hz, 2 mW, top) or single light flashes (0.5–1 mW, bottom). (J) Number of spikes fired in response to each light flash in the stimulus train depicted before and after application of 5HT. (K–L) Change in amplitude (K) and decay time constant (tau, (L) of synaptic responses before (black) and after 5HT (blue). *p<0.05, **p<0.01.

DOI: https://doi.org/10.7554/eLife.31991.002

The following figure supplements are available for figure 1:

**Figure supplement 1.** Modulation of FSI intrinsic properties by 5HT.

DOI: https://doi.org/10.7554/eLife.31991.003

**Figure supplement 2.** Dose response for 5HT.

DOI: https://doi.org/10.7554/eLife.31991.004

**Figure supplement 3.** 5HT does not change membrane potential or input resistance of SOM interneurons.

DOI: https://doi.org/10.7554/eLife.31991.005

---

unaltered by 5HT application (p=0.34 for $V_m$ and p=0.14 for $R_{in}$, *paired t-test 5HT vs. baseline*, n = 6). Overall, this indicates that FSIs respond to serotonin in a unique manner.

To determine if activation of endogenous serotonergic fibers could also increase FSI excitability in mPFC, we expressed ChR2 in serotonergic neurons in the dorsal raphe nucleus. A Cre-dependent virus (AAV5-Ef1a-DIO-ChR2-eYFP) was injected into SERT-Cre mice, which express Cre-recombinase under the promoter for the serotonin transporter (*Figure 1E*). Immunohistochemistry confirmed that both ChR2 expressing cells in the DRN and axon terminals in mPFC contained 5HT (*Figure 1E*). After waiting 5+ months for trafficking of ChR2 to prefrontal terminals, whole-cell recordings were made from prefrontal FSIs and serotonergic terminals were stimulated with rhythmic flashes of light (470 nm,~2 mW, 5 ms flashes, 10 Hz for 10 s). Endogenous 5HT release increased FSI firing rate from 10.9 ± 3.8 Hz to 29.8 ± 6.8 Hz (*Figure 1F*, p=0.035, during stimulation vs. before stimulation, paired t-test, n = 7). This increase was blocked by the 5HT2AR antagonist MDL100907 (*Figure 1G,* 1 µM, p=0.32, during stimulation vs. before stimulation, paired t-test, n = 5).

Membrane depolarization can increase excitability and, therefore, neuronal firing in response to synaptic inputs. To further investigate effects of 5HT on FSI responses to synaptic inputs (as opposed to somatic current injection), we optogenetically activated glutamatergic inputs from contralateral mPFC while recording from FSIs (*Figure 1H–I*, 470 nm, ~2 mW, 5 ms flashes, 10 flashes at 5 Hz). 5HT application increased the number of spikes elicited by each light flash (*Figure 1G*, p<0.0001 for both treatment and flash number in repeated measures two-way ANOVA, p<0.0001 for the interaction term, p<0.05 for flashes 1–8 5HT vs. baseline post-hoc comparison Bonferroni correction, n = 9).

By increasing membrane resistivity, 5HT could affect how synaptic inputs are filtered along FSI dendrites. To test this hypothesis, we delivered single light flashes at lower light power (~0.5–1 mW) and recorded excitatory post-synaptic potentials (EPSPs) before and after 5HT application (*Figure 1I* bottom). Interestingly, EPSP amplitude did not change (*Figure 1K*, p=0.120, paired t-test 5HT vs. baseline, n = 7, *Figure 1H*). However, EPSPs decayed more slowly following 5HT application (*Figure 1L*, decay tau: baseline 10.4 ± 1.2 ms, 5HT: 17.4 ± 3.5 ms, p=0.037, same analysis, n = 7).

## 5HT decreases inward-rectifying potassium channel function

5HT could modulate FSI intrinsic properties by regulating membrane ion channels. Specifically, a decrease in the overall potassium conductance, either through a direct effect on the channel or through channel internalization, would explain both of our observations: a depolarization of $V_m$ and an increase in $R_{in}$. To determine whether 5HT altered $K^+$ channel function, we made voltage clamp recordings during a membrane potential ramp from −130 mV to −50 mV over 3 s. We measured the total whole-cell current throughout the ramp before and after 5HT application (*Figure 2A* top). 5HT decreased the slope of the I-V curve (p=0.037, paired t-test 5HT vs. baseline, n = 7), indicating a decrease in membrane conductance. By subtracting the baseline I-V curve from 5HT, we calculated the I-V curve for the current modulated by 5HT (*Figure 2A* bottom). This current displayed inward rectification and reversed at −99 ± 4.7 mV, very close to the predicted reversal potential for $K^+$ in our preparation (−101 mV). Both the 5HT-mediated current (p=0.0007, KGluc +MDL100907 vs.

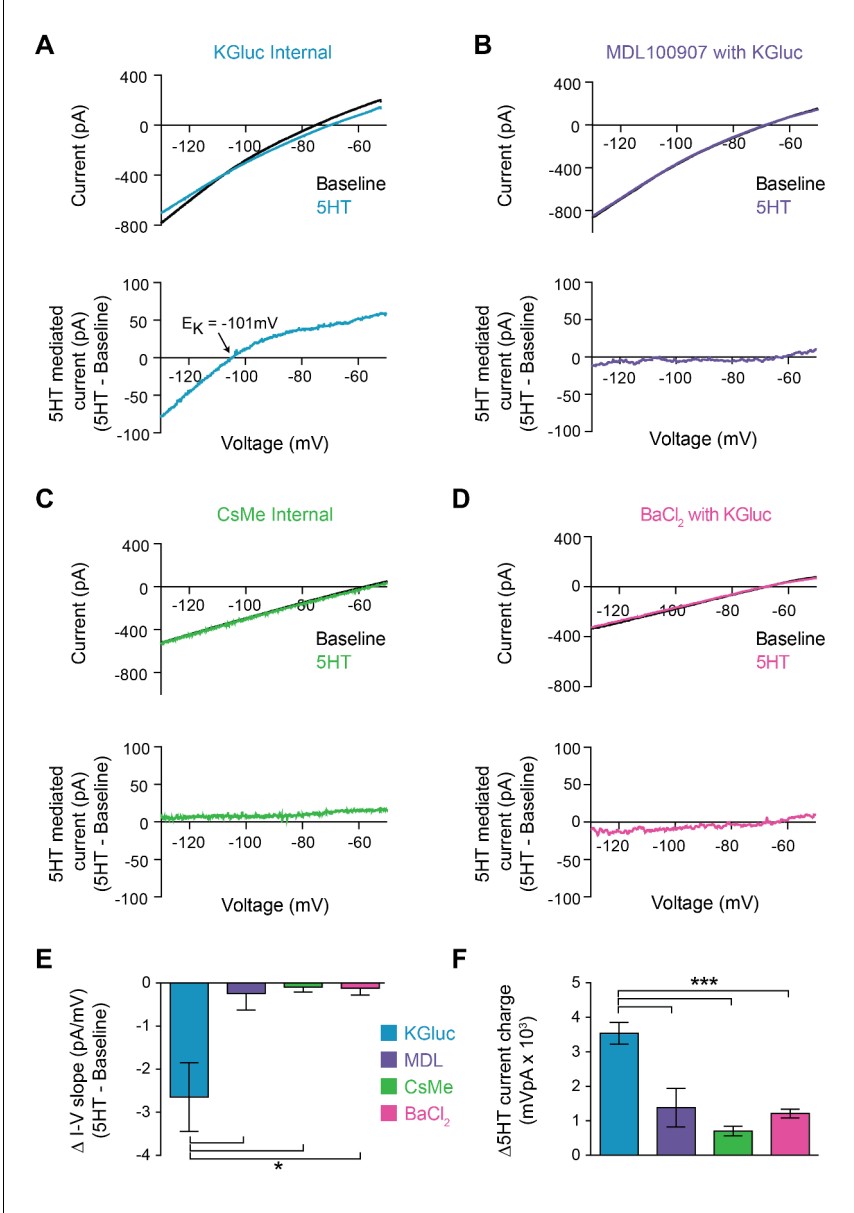

**Figure 2.** 5HT reduces conductance through inward rectifying potassium channels. (**A–B**) Top: Current recorded during a voltage ramp (3 s) from −150 mV to −50 mV before (black) and after 5HT (blue) using KGluconate in the internal solution (**A**) and with pre-application of the 5HT2A antagonist MDL100907 (1 μM). Bottom: The raw currents from the I-V curves subtracted from each other to show the current modulated by 5HT. (**C**) Top: Current in response to voltage ramp using CsMe internal solution to block K⁺ channels. Bottom: 5HT-mediated current. (**D**) Top: Current in response to same voltage ramp with barium chloride (400 μM) in the bath solution to block inward-rectifying K⁺ channels. Bottom: 5HT-mediated current. (**E**) The change in slope of the I-V curves (change in conductance) from A-D. (**F**) Quantification of charge transfer by 5HT in above conditions, calculated by taking the integral of bottom traces in A-D. *p<0.05, ***p<0.005.

DOI: https://doi.org/10.7554/eLife.31991.006

KGluc, post-hoc Tukey multiple comparison test with one-way ANOVA with p<0.0001, *Figure 2F*) and I-V curve slope change (p=0.04, same analysis, *Figure 2E*) were abolished (n = 3) with pre-application of MDL100907 (1 μM) (*Figure 2B*).

5HT-sensitive currents reversed at potassium equilibrium and were smaller in the outward direction, suggesting that 5HT modulates an inward rectifying K⁺ channel. To test this, we first repeated

voltage ramps using a cesium-based internal solution to block K$^+$ channels (**Figure 2C**). Both the 5HT-mediated current (p<0.0001, CsMe vs. KGluc, ANOVA with post-hoc Tukey multiple comparison test, **Figure 2F**) and changes in I-V curve (p=0.007, same analysis, **Figure 2E**) were absent (n = 6) in these conditions. Secondly, we switched back to a K-based internal solution and blocked inward-rectifying K$^+$ channels using Ba$^+$ (**Figure 2D**) at a concentration that fully blocks K$_{ir}$ ion flux across the entire voltage range of the ramp (400 µM, **Alagem et al., 2001**). Again, both the I-V curve changes (p<0.0001, BaCl$_2$ vs. KGluc, ANOVA with post-hoc Tukey multiple comparison test, **Figure 2E**) and the 5HT-mediated current (p=*0.01*, same analysis, **Figure 2F**) were blocked (n = 5). Thus, 5HT reduces the conductance of inward rectifying potassium channels in FSIs.

Reducing K$^+$ conductance in a compartmental model of a fast-spiking interneuron (**Figure 3A**) also increased V$_m$ and R$_{in}$ (**Figure 3B**); a reduction of g$_K$ by 60–70% across the cell was sufficient to

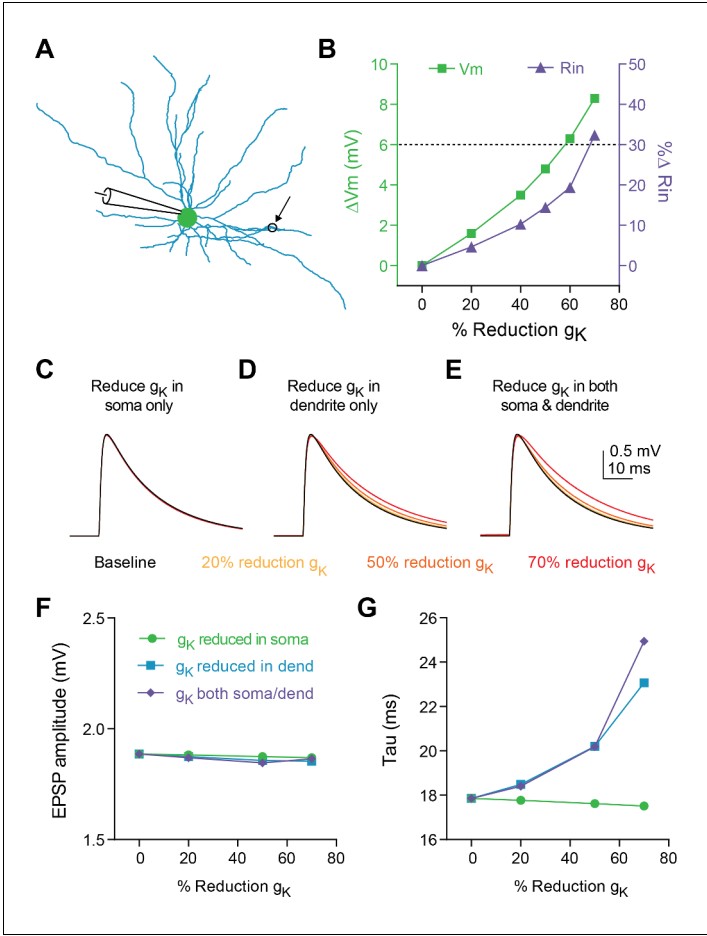

**Figure 3.** Decreasing dendritic K + conductance elicits change in tau of synaptic responses in a compartmental model. (**A**) Morphology of FSI model. Black circle represents location of synapse. (**B**) Change in membrane potential (left axis) and input resistance (right axis) in response to reducing conductance of K + channels in both the soma and dendrites of model. Dashed black line indicates average effect from experimental data on Vm and Rin in response to 5HT application. (**C–E**) Synaptic responses recorded at the model soma after reducing K + conductance by varying amounts (30–80%) only at the soma (**C**), only at the dendrites (**D**), or at both the soma and dendrites (**E**). (**F–G**) EPSP amplitude (**F**) and synaptic time constant (**G**) after reducing K + conductance in soma (green), dendrites (purple), or both (blue).

DOI: https://doi.org/10.7554/eLife.31991.007

The following figure supplement is available for figure 3:

**Figure supplement 1.** 5HT-induced dendritic depolarization reduces synaptic driving force in compartmental model.

DOI: https://doi.org/10.7554/eLife.31991.008

recapitulate our experimental data. However, increased cable filtering and prolonged synaptic decay (tau) is likely due to reduction of $K^+$ channel conductance specifically in the dendrites. Indeed, when $K^+$ channel conductance was systematically reduced in different subcompartments (*Figure 3C* soma only, *Figure 3D* dendrites only, *Figure 3E* both soma and dendrites), only dendritic reduction of $g_K$ resulted in an increase in synaptic time constant (*Figure 3G*). Interestingly, consistent with our experimental observations during optogenetic stimulation of inputs to FSIs (*Figure 1L*), no manipulation of $g_K$ affected EPSP amplitude in our model (*Figure 3F*).

This last observation can be understood using a simple computational model as follows. According to Ohm's law, increasing $R_{in}$ might be expected to elicit an increase in EPSP amplitude ($V_{syn} = I_{syn}R_{in}$). However, this could be countered by a decrease in driving force ($V_m - E_{syn}$) if the membrane depolarization from 5HT was sufficiently large. In fact, in our compartmental model, reducing the dendritic $g_K$ to mimic the experimentally observed effects of 5HT on *somatic* $V_m$ and $R_{in}$ caused a *decrease* in the EPSP amplitude and synaptic current when measured at the dendrite (90% of baseline amplitude, *Figure 3—figure supplement 1A–C*), indicating that this reduction in driving force dominates the dendritic response to synaptic stimulation. However, due to the increase in $R_{in}$, the dendritic EPSP also attenuates less as it travels toward the soma such that the EPSP amplitude, measured at the soma, is ultimately unchanged by this manipulation. As such, the effects of reducing dendritic $g_K$ on EPSP amplitude and tau should depend on the distance of the synapse from the soma (*Figure 3—figure supplement 1D*). Indeed, in our model, placing the synapse closer to the soma (<100 μm) and reducing dendritic $g_K$ to match the observed effects of 5HT on $V_m$ and $R_m$ slightly reduced the somatic EPSP amplitude (*Figure 3—figure supplement 1E*). By contrast, moving the model synapse further out along the dendrite (100–150 μm) did not affect the EPSP amplitude, presumably indicating that for a range of dendritic locations, the effects on driving force and EPSP attenuation cancel out. At distances furthest from the soma (>150 μm), somatic EPSP amplitude was increased after reducing $g_K$. We did consistently observe an increase in EPSP tau especially when the model synapse was placed far from the soma (*Figure 3—figure supplement 1F*). These results from our computational model provide simple intuition as to how 5HT can modulate the EPSP decay without affecting EPSP amplitude.

Since our modeling results suggest that 5HT acts mainly to suppress $K^+$ channels located in the dendrites, we decided to test this experimentally by delivering 5HT exclusively to the dendrites using local iontophoresis under two-photon guidance (*Figure 4A*). 5HT significantly increased FSI firing rate when applied immediately adjacent to a dendrite (*Figure 4B*, p=*0.001*, before iontophoresis vs. after iontophoresis, paired t-test, n = 9). By computationally filtering out spikes from our traces, we determined that dendritic iontophoresis of 5HT elicited a 1.19 ± 0.69 mV depolarization of the soma (*Figure 4—figure supplement 1A-B*). By applying this same filtering technique to spike trains elicited by somatic current injection, we found that a somatic depolarization of 2.63 ± 1.57 mV of FSIs was sufficient to cause an increase in firing rate comparable to that observed by dendritic iontophoresis of 5HT (*Figure 4—figure supplement 1C*). This effect disappeared when the iontophoretic pipette was withdrawn from the dendrite (*Figure 4C–D*, p=0.14, same analysis, n = 7). By placing the iontophoretic pipette at varying distances from the dendrite, we determined that 5HT only increased firing rate when the pipette was <5 μm away (space constant = 3.13 μm, *Figure 4E*). All iontophoretic sites were >30 μm from the soma (*Figure 4E*), confirming that effects are due to local action of 5HT at receptors on the dendrite and not due to diffusion of 5HT to the soma.

## 5HT enhances the temporal integration of inputs at gamma frequencies

By prolonging the decay of synaptic potentials, 5HT could enhance the integration of multiple synaptic inputs within FSI dendrites. For example, 5HT could improve summation of a second EPSP that arrives during the period of prolonged decay (~10–20 ms after the first input, *Figure 1I*). This suggests that 5HT might promote temporal summation (and spike output) in response to inputs arriving specifically in the 50 to 100 Hz range (i.e. in line with the prolonged decay), To test this, we first mimicked glutamatergic input onto FSI dendrites using two-photon flash photolysis of caged-glutamate, as this technique allows precise control of the position and timing of synaptic activity. Slices were bathed in MNI-glutamate (2.5 mM), and glutamate was released at five specific sites (~1 μm apart) on FSI dendrites (720 nm 2-photon excitation, 0.5 ms pulses, *Figure 5A*). EPSPs were recorded at the soma before and after 5HT application (applied via iontophoresis directly above the slice for 5 s). 5HT delivered in this manner produced increases in membrane potential, input

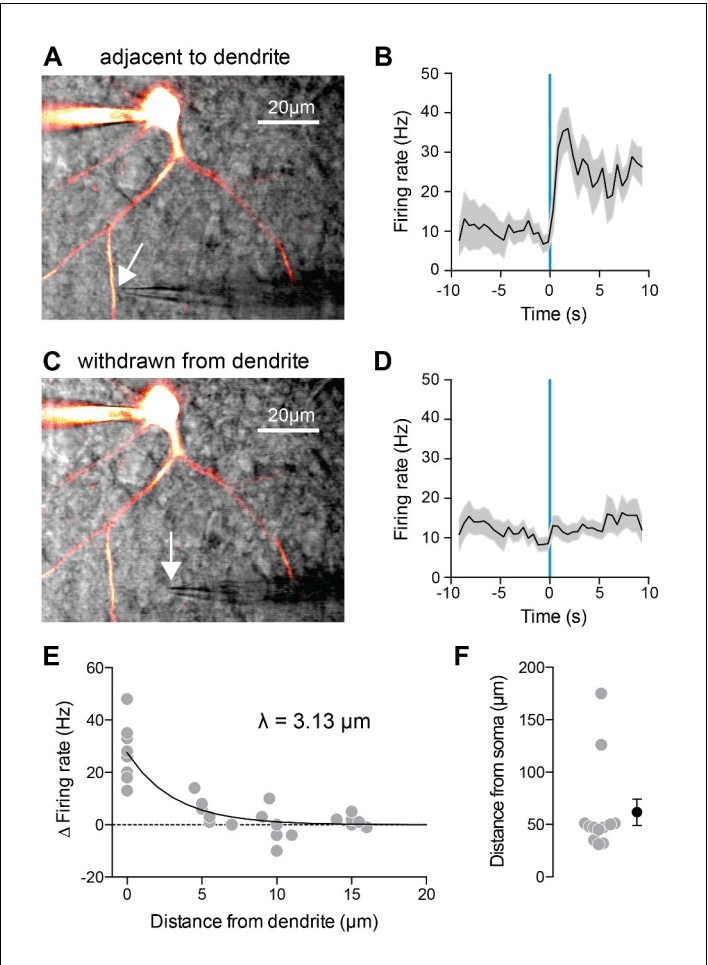

**Figure 4.** Local 5HT iontophoresis at FSI dendrites increases FSI firing. (**A,C**) Experimental design: Neurons were patched and filled with Alexa-488. 5HT was applied locally to the dendrite using iontophoresis (50 ms), while FSIs were injected with a small amount of depolarizing current to elicit spiking. DIC and overlaid fluorescent images with iontophoretic pipette adjacent to (**A**) or withdrawn from (**C**) dendrite. (**B,D**) Firing rate in response to current injection with local 5HT iontophoresis adjacent to dendrite (**B**) or withdrawn from dendrite (**D**). (**E**) Change in firing rate with iontophoresis (FR 1 s before ionto pulse subtracted from FR after ionto pulse) at different distances from the dendrite. Solid black line in exponential fit to data. Space constant is 3.13 μm. (**F**) Distances of iontophoretic sites from the soma.

DOI: https://doi.org/10.7554/eLife.31991.009

The following figure supplement is available for figure 4:

**Figure supplement 1.** Dendritic 5HT iontophoresis depolarizes neuron sufficiently to induce observed change in firing rate.

DOI: https://doi.org/10.7554/eLife.31991.010

resistance, and firing rate comparable to bath application (***Figure 5—figure supplement 1***). Consistent with ChR2 synaptic stimulation experiments (***Figure 1H–L***) and compartmental modeling (***Figure 3***), 5HT did not affect single EPSP amplitude (***Figure 5C***, p=0.45, normalized change in amplitude: Post/Pre, one-sample t-test vs. 1, n = 39 dendrites, 25 cells). However, 5HT did increase the time constant of the EPSPs from 14.7 ± 0.6 ms to 18.6 ± 1.0 ms (***Figure 5D***, p=0.0002, change in tau: Post – Pre, one-sample t-test vs. 0, n = 39 dendrites, 25 cells). As with ChR2 synaptic stimulation, changes in EPSP decay peaked 10–20 ms after stimulation (***Figure 5E***), suggesting that the summation of synaptic inputs arriving within this time window may be preferentially enhanced by 5HT.

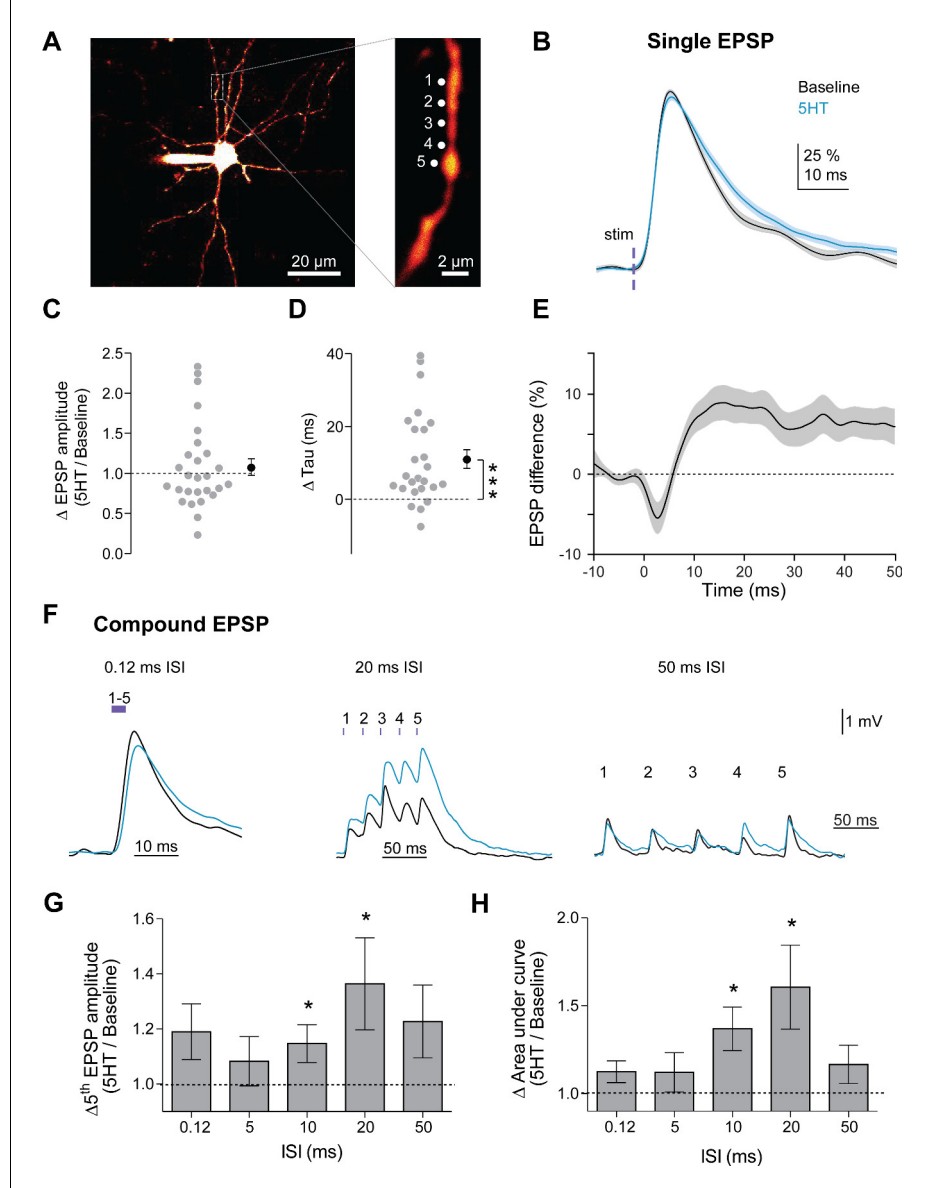

**Figure 5.** 5HT promotes integration of synaptic inputs in a frequency-specific manner. (A) Experimental design: Slices were bathed in a caged glutamate compound (MNI-Glutamate 2.5 mM) that is only biochemically active with photolysis. Glutamate was uncaged at five locations (1 μm apart) on a dendrite individually and then at all five together with varying interstimulus intervals (0.12, 5, 10, 20, 50 ms). (B) Amplitude-normalized EPSP in response to single uncaging events before (black) and after application of 5HT (blue). (C) Ratio of EPSP amplitudes (5HT/baseline) for single uncaging events (averaged per dendrite). Dotted line indicates no change. (D) Change in synaptic decay time constant (tau) of single uncaging events (averaged per dendrite) before and after 5HT. Dotted line indicates no change. (E) Difference of 5HT and baseline EPSP traces in B. (F) Example compound EPSPs in response to uncaging at all five dendritic locations at varying interstimulus intervals (0.12, 20, 50 ms). (G–H) Ratio of fifth EPSP amplitude (G, 5HT/Baseline) and charge transfer (H, 5HT/Baseline integral) for different interstimulus intervals. *$p < 0.05$, **$p < 0.01$.

DOI: https://doi.org/10.7554/eLife.31991.011

The following figure supplement is available for figure 5:

**Figure supplement 1.** Broad 5HT iontophoresis over the slice produces typical 5HT effects.
DOI: https://doi.org/10.7554/eLife.31991.012

To examine frequency-specific effects of 5HT on temporal summation, we uncaged glutamate at all five sites using varying inter-stimulus intervals (0.12, 5, 10, 20, 50 ms, *Figure 5F*), and measured the amplitude of the last EPSP as well as the total integrated EPSP area. As predicted, 5HT significantly promoted summation, specifically at the 10 ms (p=*0.05*, change in fifth EPSP amplitude: 5HT/ Baseline, one-sample t-test vs. 1, n = 26 dendrites, 18 cells; p=*0.006*, change in area under curve: 5HT/Baseline, one-sample t-test vs. 1, n = 26 dendrites, 18 cells, *Figure 5H*) and 20 ms intervals (p<0.05, change in fifth EPSP amplitude, same analysis as above, n = 29 dendrites, 20 cells, *Figure 5G*; p=0.0171, change in area under curve, same analysis as above, n = 29 dendrites, 20 cells, *Figure 5H*). Thus, by slowing the decay of synaptic potentials, 5HT promotes summation of gamma frequency inputs.

## Computational models reproduce frequency-specific enhancement of summation by 5HT

To determine if the change in synaptic decay could fully account for the observed differences in temporal summation, we first used a simple algebraic model to explore the effect of changing the synaptic time constant independent of changes in membrane potential and resistivity. After creating an EPSP template using a double exponential equation, the decay time constant was altered to mimic the effect of 5HT ($tau_{baseline}$ = 15 ms, $tau_{5HT}$ = 23ms) and five template EPSPs were convolved at varying intervals (0–50 ms). Consistent with experimental data, changing the tau of the EPSP decay promoted summation of inputs at 10–20 ms intervals more than other frequencies (*Figure 6B*).

We then implemented this change in tau in the FSI compartmental model by reducing $g_k$ in all compartments. To simulate our two-photon uncaging experiment, five model synapses along a single dendrite (1 µm apart, *Figure 6C*) were stimulated using varying interstimulus intervals (*Figure 6E*). A 70% reduction in $g_K$ throughout the neuron changed $V_m$, $R_{in}$, and tau by an amount comparable to the experimental effect of 5HT (*Figure 6D*). Furthermore, in the model, the reduction in $g_K$ and resultant change in tau could reproduce the frequency-specific enhancement of summation observed in our uncaging experiments. Specifically, inputs arriving within 10–20 ms were summated preferentially, as compared to higher (0–5 ms ISI) or lower (50 ms ISI) frequencies. Preferential summation at 10–20 ms ISIs was observed even when we varied synaptic strength, synapse placement on the dendritic arbor, or number of synapses recruited (*Figure 6—figure supplement 1*). While the same general effect was observed when all synapses were clustered onto the soma, we found that peak amplitudes were attenuated, likely due to reductions in driving force from membrane depolarization (*Figure 6—figure supplement 1B-C*). Next, we included random background synaptic noise to reduce $R_{in}$ and more closely simulate in vivo conditions. For these experiments, we included randomly fluctuating noise conductances that modeled both excitatory and inhibitory conductances on a subset of dendrites and stimulated synapses at baseline and after reducing K + conductance by 70%. As expected, inclusion of noise reduced the measured input resistance (*Figure 6—figure supplement 2C*). But even when $R_{in}$ was reduced up to 33%, summation remained tuned to enhancements for 10–20 ms (*Figure 6—figure supplement 2D*). These values encompass those measured in vivo (47 MΩ; *Pala and Petersen, 2015*), suggesting that serotonin enhances frequency-specific summation over a broad range of background activity levels.

## FSIs preferentially spike in response to gamma frequency inputs with 5HT

Preferential summation of gamma-frequency inputs could, in turn, promote gamma-specific FSI output. To isolate the intrinsic effect of 5HT in FSIs and avoid any potential off target effects of 5HT modulation at other loci within the prefrontal microcircuit, we took a chemogenetic approach, expressing the Gq-coupled designer receptor hM3D (AAV-DJ-Ef1a-DIO-h3MD(Gq)-mCherry), which is activated by the exogenous ligand clozapine-N-oxide (CNO), or a control fluorophore virus (AAV-DJ-Ef1a-DIO-mCherry) in FSIs using PV-Cre[+/-] mice. CNO had no effect on $V_m$ or Rin in FSIs expressing only the control fluorophore (*Figure 7—figure supplement 1*). However, in FSIs expressing the Gq-DREADD receptor, CNO application increased $V_m$ (*Figure 7C*; p<0.0001, paired t-test CNO vs. baseline, n = 7), $R_{in}$ (*Figure 7D*; p=0.0004, paired t-test CNO vs. baseline), and spiking (*Figure 7B*) in FSIs, and produced similar changes in $K_{ir}$ function (*Figure 7E*), suggesting that hM3D receptors

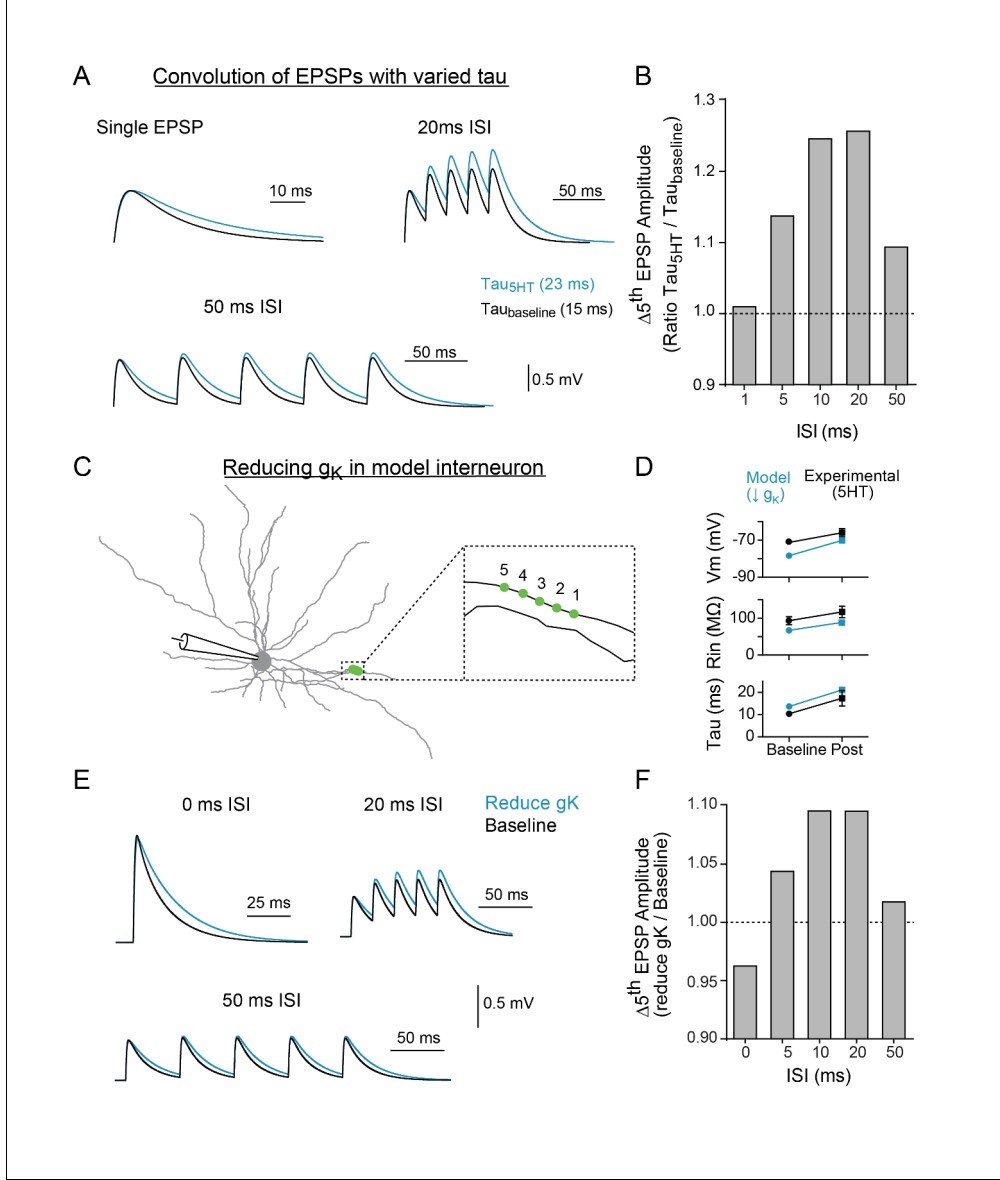

**Figure 6.** Modeling indicates that changing tau and reducing K + conductance can modulate temporal summation. (**A**) Experimental design: Single EPSPs were modeled using a double exponential with two different decay constants (tau$_{baseline}$ = 15 ms, tau$_{5HT}$ = 23 ms) to match the change in tau observed with 5HT application. These template EPSPs were convolved five times with varying intervals (ISI). Example traces shown here of single EPSP, 20 ms ISI, and 50 ms ISI with baseline tau (black) and 5HT tau (blue). (**B**) The ratio of amplitude of the fifth EPSP (slow tau/fast tau) as a function of interval. (**C**) Morphology of FSI model. Purple circles represent location of synapses. (**D**) Comparisons of model (blue) and experimental (black) intrinsic properties at baseline (circles) and after manipulations (blue = model: reducing g$_K$ by 70%, black = experimental: application of 5HT). (**E**) Synapses in model were stimulated at variable interstimulus intervals (ISI). Example traces for compound EPSPs with 0 ms, 20 ms, and 50 ms ISIs at baseline (black) and after reducing g$_K$ by 70% (blue). (**F**) The ratio of amplitude of the fifth EPSP (reduced g$_K$/baseline) as a function of ISI.
DOI: https://doi.org/10.7554/eLife.31991.013

The following figure supplements are available for figure 6:

**Figure supplement 1.** Model is robust to changes in synaptic parameters.
DOI: https://doi.org/10.7554/eLife.31991.014

**Figure supplement 2.** Background synaptic noise does not change summation enhancement.
DOI: https://doi.org/10.7554/eLife.31991.015

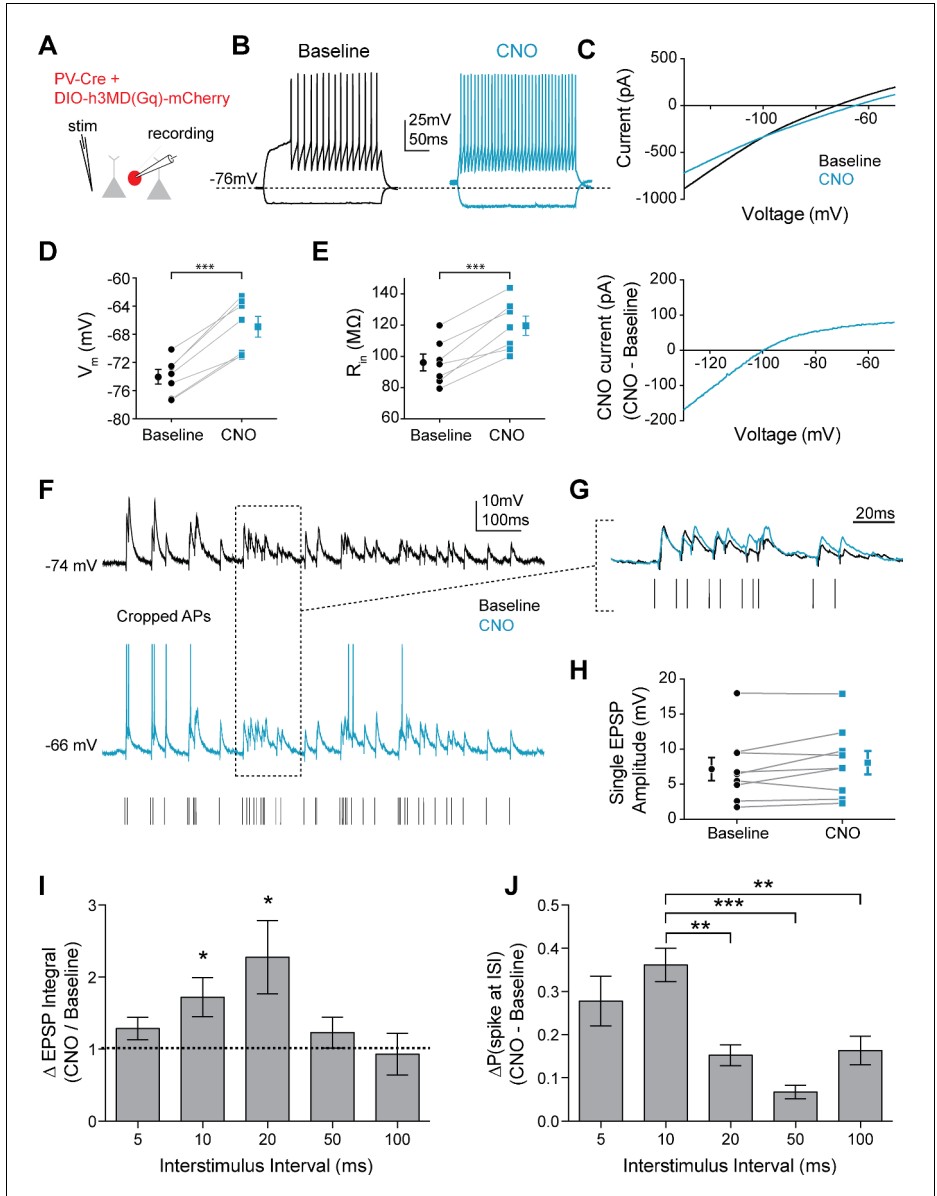

**Figure 7.** Mimicking 5HT effects elicits preferential firing to gamma frequency inputs in FSIs. (**A**) Experimental design: The Gq-coupled Designer Receptor Exclusively Activated by Designer Drugs (DREADD) was expressed specifically in FSIs using a Cre-dependent virus injected into PV-Cre mice. FSIs were identified with fluorescence for patching. (**B**) Example FSI responses to hyperpolarizing and depolarizing current steps at baseline (black) and after application of CNO to activate the Gq-DREADD (1 μM, blue). (**C**) Top: Current recorded during a voltage ramp (3 s) from −150 mV to −50 mV before (black) and after CNO (blue). Bottom: The raw currents from the I-V curves subtracted from each other to show the current modulated by CNO. (**D–E**) Changes in membrane potential (**D**) and input resistance (**E**) before and after CNO. (**F**) Experimental design: A stimulating electrode was placed in the tissue within 100 μm of the recorded FSI and a 2 s train of randomly distributed stimulating current pulses (200 μs) with varied interstimulus intervals (ISIs = 5, 10, 20, 50, 100 ms) was delivered. Example FSI responses to stimulus train at baseline (black) and after application of CNO (blue). (**G**) Expanded view of subthreshold responses indicated by dotted box in F. (**H**) Change in single EPSP amplitude with CNO. (**I**) Normalized change in EPSP integral (CNO/Baseline) with CNO application for different ISIs. (**J**) Change in the percentage of all spikes occurring at each ISI after CNO. *p<0.05, **p<0.01 ***p<0.005.
DOI: https://doi.org/10.7554/eLife.31991.016

The following figure supplements are available for figure 7:

**Figure supplement 1.** CNO has no effect on FSIs not expressing DREADD.

*Figure 7 continued on next page*

*Figure 7 continued*

DOI: https://doi.org/10.7554/eLife.31991.017

**Figure supplement 2.** 5HT2A agonist increases probability of FSI firing in response to gamma frequency inputs.
DOI: https://doi.org/10.7554/eLife.31991.018

can co-opt signaling pathways downstream of 5HT2A receptors and mimic the effects of 5HT on FSI physiology.

We delivered a train of randomly distributed electrical stimuli (*Figure 7F–G*) using predefined ISIs (5, 10, 20, 50, 100 ms) through a local stimulating electrode. Again, we observed no measurable change in single EPSP amplitude (p=0.13, paired t-test CNO vs. baseline, n = 7). Interestingly, CNO application mainly enhanced EPSPs following ISIs of 10 or 20 ms (p=*0.04* for 10 ms and p=0.046 for 20 ms, normalized change in EPSP integral: CNO/baseline, one sample t-test vs. 1, n = 7, *Figure 7I*). The increased summation at these frequencies translated into an increase in the probability of firing in response to inputs after 10 ms ISIs. Specifically, CNO increased the percentage of spikes that occurred after stimulation with 10 ms ISI compared to other frequencies (p<0.0001 for treatment in ANOVA, p=0.002 for 10 ms vs. 20 ms, p<0.0001 for 10 ms vs. 50 ms, p=0.004 for 10 ms vs. 100 ms, post-hoc comparison with Tukey's correction, *Figure 7I*). While the Gq-DREADD activates the same downstream signaling cascades, it is not clear if its localization is similar to that of endogenous 5HT2A receptors. Therefore, we performed similar experiments using a selective 5HT2A agonist α-methyl-5HT (30 μM). Effects on FSI $V_m$ (p=0.002, paired t-test baseline vs. α-methyl-5HT, n = 5) and $R_{in}$ (p=0.01, paired t-test baseline vs. α-methyl-5HT, n = 5) were similar with α-methyl-5HT as with endogenous 5HT and Gq-DREADD activation with CNO (*Figure 7—figure supplement 1*; *Figure 7—figure supplement 2B–C*). As in other experiments, we found that α-methyl-5HT did not change the amplitude (p=*0.96*, paired t-test baseline vs. α-methyl-5HT, n = 5, *Figure 7—figure supplement 2*), but did increase the decay time (p=0.03, paired t-test baseline vs. α-methyl-5HT, n = 5, *Figure 7—figure supplement 2E*) of subthreshold EPSPs. Furthermore, we replicated our earlier findings and found that 5HT2AR agonism increased the percentage of spikes that occurred after stimulation with 10 ms and 20 ms ISIs compared to other frequencies (p<0.001 for treatment in ANOVA, *p*=0.002 for 10 ms vs. 50 ms, p=0.002 for 10 ms vs. 100 ms, p=0.01 for 5 ms vs. 20 ms, p=0.002 for 20 ms vs. 50 ms and 100 ms post-hoc comparison with Tukey's correction, 6 cells, *Figure 7—figure supplement 2D*). Thus, serotonergic signaling in FSIs seems to specifically enhance the ability of these cells to respond to gamma-frequency input.

Due to heavy reciprocal connectivity, FSIs are able to entrain the synchronous firing of neighboring interneurons and pyramidal cells, and an increased probability of firing in response to gamma frequency inputs in FSIs could promote gamma frequency inhibition in the prefrontal cortical microcircuit. Therefore, we recorded spontaneous inhibitory postsynaptic currents (IPSCs) in pyramidal neurons in an 'active' slice preparation (2 μM carbachol) at baseline and after Gq-DREADD activation in FSIs with CNO (*Figure 8A–B*). We found that CNO dramatically increased the number of IPSCs recorded (p=*0.04* paired t-test CNO vs. baseline, 5 cells, *Figure 8C*), indicating an overall increase in inhibition. Interestingly, we also found that the frequency of IPSCs shifted toward inter-event intervals of 10–20 ms (*Figure 8D*), corresponding to an increase in the probability of gamma frequency inhibitory events (*Figure 8E*). Thus, we conclude that 5HT increases gamma frequency inhibition in the prefrontal network.

## Discussion

Here, we provide a detailed examination of how changes to passive membrane properties by a neuromodulator alter temporal integration by a neuron. We find that by closing potassium channels, serotonin not only increases the excitability of FSIs, but also promotes synaptic integration in a frequency-specific manner, leading to preferential enhancement of responses to gamma frequency inputs, both in terms of EPSP summation, spiking, and network inhibition.

### 5HT increases FSI excitability by altering intrinsic properties

We described the detailed cellular mechanism through which 5HT modulates FSI activity in the mPFC. Previous slice physiology studies showed that 5HT or a 5HT2A agonist could increase the

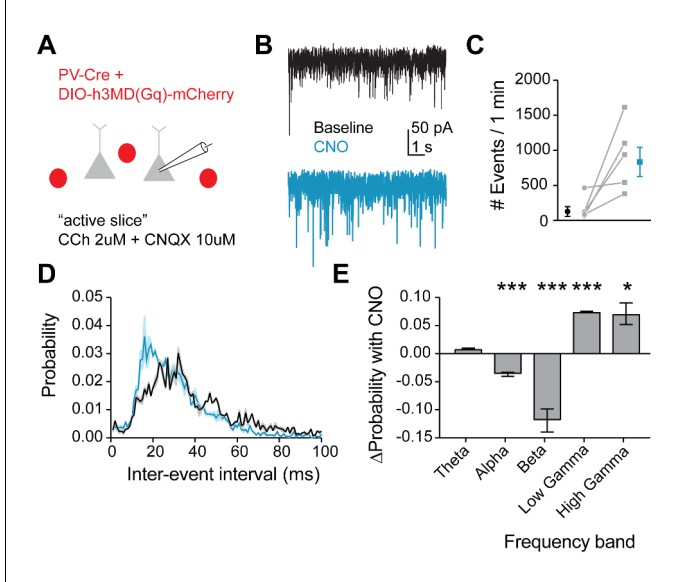

**Figure 8.** Mimicking 5HT effects in FSIs produces gamma frequency events in downstream pyramidal neurons. (A) Experimental design: The Gq-DREADD was expressed specifically in FSIs using a cre-dependent virus injected into PV-Cre mice. Prefrontal slices were bathed in carbachol (2 μM) to induce spontaneous background synaptic activity. Non-fluorescent pyramidal neurons were chosen for patching using a high chloride internal solution to elicit inward IPSCs. CNQX (10 μM) was included in the bath to block AMPA currents. (B) Example traces of spontaneous IPSCs recorded at baseline (black) and after wash-in of CNO (blue). (C) Total number of IPSC events in 1 min at baseline and after CNO application. (D) Probability distribution of inter-event intervals for IPSCs recorded at baseline and after CNO. (E) Change in probability of inter-event intervals of different frequency bands. Theta = 48 Hz, alpha = 8–12 Hz, beta = 13–29 Hz, low gamma = 39–59 Hz, high gamma = 61–100 Hz. *p<0.05, **p<0.01 ***p<0.005.
DOI: https://doi.org/10.7554/eLife.31991.019

frequency of spontaneous inhibitory post-synaptic currents (sIPSCs) recorded in pyramidal neurons (*Weber and Andrade, 2010*; *Zhou and Hablitz, 1999*) and that 5HT increased firing of FSIs (*Weber and Andrade, 2010*; *Zhong and Yan, 2011*). Here, we showed that 5HT increased FSI input resistance, membrane voltage, and AP excitability in response to both somatic current injection (*Figure 1B*, *Figure 1—figure supplement 1D*) and optogenetically evoked synaptic input (*Figure 1H–L*). These changes in intrinsic properties reflect the reduction of an inward-rectifying potassium conductance in FSIs (*Figure 2*). These data may provide mechanistic insight into a previously observed increase in presynaptic facilitation of inhibitory glycinergic synapses (*Mintz et al., 1989*). Our results are in contrast to one study that found that 5HT differentially increases or decreases firing of distinct subpopulations of FSIs in vivo (*Puig et al., 2010*). However, these experiments were performed in a different species (rat) and in a more dorsolateral brain region (M2). Furthermore, systemic 5HT antagonists in that study could have influenced other cells in the network such that the observed effects may not reflect direct actions on FSIs.

## 5HT promotes temporal summation of inputs arriving at gamma frequency by prolonging EPSP decay

Compared to neighboring pyramidal cells, which can respond to synaptic input with the generation of dendritic superlinearities (*Stuart and Spruston, 2015*), FSI dendrites tend to function more as passive filters (*Abrahamsson et al., 2012*). As such, changes in passive membrane properties can have significant effects on integration in FSIs. Here, 5HT-mediated suppression of dendritic $K_{ir}$ conductances prolonged the decay of synaptic potentials without changing EPSP amplitude (*Figure 1K–L*, *Figure 5A–E*). This provides a mechanism for promoting the summation of inputs arriving during the period of prolonged decay, and specifically enhancing high gamma frequency (50–100 Hz) inputs, as compared to input at other frequencies. FSIs also exhibit subthreshold

resonance around 30 Hz (*Bracci et al., 2003*; *Fellous et al., 2001*; *Pike et al., 2000*) and network models show that this membrane resonance contributes to network gamma oscillations (*Moca et al., 2014*). Furthermore, FSIs show enhanced firing in response to gamma frequency modulation of sinusoidal current injected into the soma (*Pike et al., 2000*). However, no studies have explored whether synaptic integration in FSIs also favors gamma frequency inputs. Here, we show that the enhanced temporal integration of gamma frequency inputs elicited by 5HT also translates to a greater probability of FSI spiking by using a Gq-DREADD expressed exclusively in PV cells (*Figure 7F–I*). By enhancing summation of inputs at these frequencies, 5HT could further enhance an intrinsic preference for FSIs to fire at gamma frequencies, and thus regulate the power of gamma oscillations.

## Implications for serotonergic regulation of prefrontal circuit activity

FSIs play a critical role in shaping cortical circuit activity. With abundant reciprocal connections and highly divergent synapses onto principal pyramidal cells, FSIs are able to precisely control the timing of spike discharges of large populations of cortical neurons (*Bartos et al., 2007*; *McBain and Fisahn, 2001*; *Tamás et al., 2000*). These properties endow FSIs with the ability to orchestrate network oscillations (*Cardin et al., 2009*; *Sohal et al., 2009*), specifically in the gamma frequency range, which have been suggested to be important for information encoding (*Buzsáki and Chrobak, 1995*; *Buzsáki and Wang, 2012*). A previous study (*Puig et al., 2010*) found that electrical stimulation of the dorsal raphe of rat both increased and decreased the activity of distinct populations of FSIs in secondary motor cortex via 5HT2A and 5HT1A receptors, respectively. Furthermore, this group found that blocking 5HT2ARs decreased the power of gamma oscillations, suggesting that 5HT2AR activation could contribute to increases in gamma oscillatory power. In our preparation, we did not find distinct subpopulations of FSIs with differential responses to 5HT; all recorded FSIs displayed an increase in membrane potential, input resistance, and excitability. Therefore, by regulating the temporal summation of inputs to FSIs and resulting FSI spiking, serotonergic actions on FSI dendrites may provide a substrate for regulating the power or frequency of network oscillatory activity, thereby enhancing information transfer to downstream structures (*Sohal et al., 2009*). Future studies should examine the effect of 5HT on FSI activity and gamma oscillations in vivo.

Serotonin plays a complex role in prefrontal circuits and has been implicated in a wide array of prefrontal cognitive tasks from rule shifting (*Clarke et al., 2007*, *2004*; *Baker et al., 2011*) and executive control (*Koot et al., 2012*; *Carli et al., 2006*) to working memory (*Williams et al., 2002* ) and social cognition (*Passamonti et al., 2012*). While we focus here on the role of 5HT on FSIs, 5HT receptors are also expressed on pyramidal neurons (*Araneda and Andrade, 1991*; *Avesar and Gulledge, 2012*) and 5HT3a-expressing interneurons, including VIP, CCK, and others (*Puig and Gulledge, 2011*). Thus, it will be interesting in future studies to determine whether serotonergic regulation of one or all these populations is important for its behavioral effects.

## Clinical relevance

Prefrontal dysfunction is etiological to many major psychiatric disorders, including schizophrenia and depression (*Drevets et al., 2008*). Moreover, current treatments for these disorders often target serotonergic transmission. Selective serotonin reuptake inhibitors (SSRIs) remain the most commonly used treatments for depression (*Risch and Nemeroff, 1992*; *Willner, 1985*) and second-generation antipsychotics used in schizophrenia block the 5HT2A receptor with high affinity (*Meltzer et al., 2003*; *Meltzer and Massey, 2011*). Classic hallucinogens such as lysergic acid diethylamide (LSD) activate the 5HT2A receptor (*Titeler et al., 1988*), implicating it in psychosis. Additionally, patients with schizophrenia show lower levels of the 5HT2A receptor in PFC (*Arora et al., 1991*; *Selvaraj et al., 2014*).

Impairments in FSI function and gamma rhythms may also be involved in the pathophysiolology of schizophrenia (*Gonzalez-Burgos et al., 2010*; *Gonzalez-Burgos et al., 2015*; *Lewis et al., 2005*; *Uhlhaas and Singer, 2010*). Previous work from our group showed that optogenetic activation of FSIs at gamma frequency is able to rescue impairments in cognitive flexibility in mice that model key aspects of schizophrenia (*Cho et al., 2015*). Furthermore, treatment with antipsychotic drugs that act on 5HT receptors can reduce gamma power (*Schulz et al., 2012*). The precise relationship between 5HT, prefrontal FSIs, and schizophrenia is still unclear. However, the findings of this study

may contribute to understanding clinical actions of second-generation antipsychotics through their actions on prefrontal FSIs.

# Materials and methods

## Key resources table

| Reagent type (species) or resource | Designation | Source or reference | Identifiers | Additional information |
|---|---|---|---|---|
| genetic reagent (*M. musculus*) | PV-Cre | Jackson Laboratory | Stock#:017320 | |
| genetic reagent (*M. musculus*) | Ai14 | Jackson Laboratory | Stock#:007914 | |
| genetic reagent (*M. musculus*) | SERT-Cre | Jackson Laboratory | Stock#:014554 | |
| transfected construct (virus) | AAV5-Ef1-DIO-ChR2-eYFP | UNC Vector Core | AAV5-Ef1a-DIO-hChR2 (H134R)-EYFP-WPRE-pA | |
| transfected construct (virus) | AAV5-CaMKII-ChR2-eYFP | UNC Vector Core | AAV5-CaMKIIa-hChR2 (H134R)-EYFP | |
| transfected construct (virus) | AAV-DJ-Ef1a-DIO-hM3D (Gq)-mCherry | Stanford Vector Core | GVVC-AAV-130 | |
| transfected construct (virus) | AAV-DJ-Ef1a-mCherry | Stanford Vector Core | GVVC-AAV-14 | |
| chemical compound, drug | DL-AP5 | Tocris | Catalog#:3693 | |
| chemical compound, drug | CNQX | Tocris | Catalog#:1045 | |
| chemical compound, drug | Gabazine | Tocris | Catalog#:1262 | |
| chemical compound, drug | 5HT | Tocris | Catalog#:3457 | |
| chemical compound, drug | MDL100907 | Tocris | Catalog#:4173 | |
| chemical compound, drug | $\alpha$−methyl−5HT | Tocris | Catalog#:0557 | |
| chemical compound, drug | Carbachol | Tocris | Catalog#:2810 | |
| chemical compound, drug | MNI-Glutamate | Tocris | Catalog#:1490 | |
| antibody (rabbit) | Rabbit anti-5HT | Immunostar | Catalog#:20080 | 1:500 |
| antibody (mouse) | mouse anti-GFP | Invitrogen | Catalog#:A11120 | 1:500 |
| antibody (goat) | Alexa 405 goat anti-rabbit | Invitrogen | Catalog#:A31556 | 1:250 |
| antibody (goat) | Alexa 488 goat anti-mouse | Invitrogen | Catalog#:A11029 | 1:250 |

## Electrophysiology

Coronal brain slices (250 µm) including medial prefrontal cortex were made from adult mice aging 8 weeks or older. We used the following transgenic mouse lines: PV-Cre (RRID: IMSR_JAX:008069), PV-Cre::Ai14 (RRID:MGI:2176738), and SERT-Cre (RRID: IMSR_JAX:014554). All experiments were conducted in accordance with procedures established by the Institutional Animal Care and Use Committee and Laboratory Animal Resource Center at the University of California, San Francisco. Slicing solution was chilled to 4°C and contained (in mM): 234 sucrose, 26 $NaHCO_3$, 11 glucose, 10 $MgSO_4$, 2.5 KCl, 1.25 $NaH_2PO_4$, 0.5 $CaCl_2$, bubbled with 5% $CO_2$/95% $O_2$. Slices were incubated in artificial cerebrospinal fluid (aCSF) at 32°C for 30 min and then at room temperature until recording. aCSF contained (in mM): 123 NaCl, 26 $NaHCO_3$, 11 glucose, 3 KCl, 2 $CaCl_2$, 1.25 $NaH_2PO_4$, 1 $MgCl_2$, also bubbled with 5% $CO_2$/95% $O_2$.

Neurons were visualized using differential interference contrast or DODT contrast microscopy on an upright microscope (Olympus, Burlingame, CA). Recordings were made using a Multiclamp 700B (Molecular Devices, Sunnyvale, CA) amplifier and acquired with either pClamp or IgorPro (iontophoresis and uncaging experiments). Patch pipettes (2–5 MΩ tip resistance) were filled with the following (in mM): 130 KGluconate, 10 KCl, 10 HEPES, 10 EGTA, 2 $MgCl_2$, 2 MgATP, 0.3 $Na_3$GTP. For some voltage clamp experiments, a cesium based internal solution was used that contained (in mM): 130 $CsCH_4O_3S$, 4 NaCl, 2 $MgCl_2$, 10 EGTA, 10 HEPES, 2 MgATP, 0.5 $Na_3$GTP. For recordings of inhibitory postsynaptic currents (IPSCs) in pyramidal neurons, a high -chloride internal solution was used that contained (in mM): 120 CsCl, 15 $CsMeSO_4$, 8 NaCl, 0.5 EGTA, 10 HEPES, 2 MgATP, 0.5 $Na_3$GTP. All recordings were made at 32–34°C. Series resistance was compensated in all current

clamp experiments and monitored throughout recordings. Recordings were discarded if Rs changed by >25%.

Fast-spiking interneurons were identified by fluorescent visualization of td-Tomato (PV-Cre::Ai14 mice) or mCherry expressed by Cre-dependent viral injection (AAV5-hSyn-DIO-hM3D(Gq)-mCherry or AAV-DJ-Ef1a-DIO-mCherry, PV-Cre mice) or mCherry driven by the Dlxi12b enhancer (AAV5-Dlxi12b-mCherry, SERT-Cre mice).

All bath-applied drugs (Tocris, Minneapolis, MN) were dissolved in water (3, 15, 30, or 100 μM 5HT, 10 μM CNQX, 100 μM DL-AP5, 10 μM gabazine, 1 μM CNO, 30 μM α-methyl-5HT, 2 μM carbachol) or DMSO (1 μM MDL100907) before being diluted in aCSF. MNI-Glutamate (2.5 mM) was dissolved directly in aCSF as powder. Alexa 488 (Invitrogen) was dissolved in water and then diluted to 10 μM in internal solution. For experiments including iontophoresis, the 5HT or vehicle solutions were made to pH = 4.5 using 10N HCl. The experimenter was not blind to pharmacological treatment.

## Viral injection for expression of ChR2 or fluorescent reporter

Viral injections were performed using standard mouse stereotactical methods. Mice were anesthetized for the duration of the surgery using isofluorane gas. After cleaning, an incision was made in the scalp, the skull was leveled, and small burr holes were drilled over the brain region of interest using a dental drill. Virus was injected through the burr holes using a microinjector (WPI, Sarasota, FL) at a speed of 150 nL/min and the scalp was closed using sutures or tissue adhesive (3M, St. Paul, MN).

For expression of ChR2 in serotonergic neurons, we injected a Cre-dependent ChR2 virus (AAV5-Ef1a-DIO-ChR2-eYFP, 1 μL) into the dorsal raphe of SERT-Cre mice (>p40) and waited 5–15 months for trafficking of ChR2 to the axon terminals in mPFC. In these mice, we additionally injected an AAV-Dlxi12b-mCherry virus (750 nL) into the mPFC one month before patching to label interneurons for easy identification.

For stimulation of synaptic inputs into mPFC using ChR2, we injected a ChR2 virus driven by the CaMKII promoter (AAV5-CaMKII-ChR2-eYFP, 750 nL) unilaterally into PV-Cre::Ai14 mice and patched FSIs in the opposite hemisphere after waiting 4–5 weeks for expression.

For DREADD activation of fast-spiking interneurons, we injected a Cre-dependent virus expressing the Gq-DREADD (AAV-DJ-Ef1a-DIO-hM3D(Gq)-mCherry, 750 nL) or a control fluorophore (AAV-DJ-Ef1a-DIO-mCherry, 750 nL) into PV-Cre[+/-] mice and patched from fluorescent cells after waiting 5 weeks for expression.

Dorsal raphe injection coordinates were A/P=−4.55, M/L = 0.0, D/V = −3.0. mPFC injection coordinates were A/P=1.7, M/L = ±0.3, D/V = −2.75

## ChR2 stimulation

We stimulated ChR2 in terminals using 5 ms flashes of light generated by a Lambda DG-4 (Sutter Instruments) high-speed optical switch with a 300 W Xenon lamp delivered through a 470 nm excitation filter. For stimulation of 5HT terminals, light flashes were delivered at 10 Hz for 10 s through a 40x objective. For stimulation of ChR2 from contralateral PFC, we delivered a train of 10 light flashes at 5 Hz.

## Electrical stimulation

mPFC synapses were stimulated using an IsoFLEX stimulator (AMPI, Israel, 200 μs pulse duration) via a bipolar glass stimulation electrode (Sutter, Novato, CA) placed within 100 μm of the patched cell. A 2-s stimulus train was delivered where stimulus pulses of varying interstimulus intervals (5, 10, 20, 50, 100 ms) were randomly distributed in the train.

## Two-photon imaging, glutamate uncaging, and 5HT iontophoresis

Neurons were visualized using a two-photon imaging system (Bruker, Middleton, WI) powered by two femtosecond lasers (Coherent, Ultra II) as previously described (Bender and Trussell, 2009). Internal solution was supplemented with 10 μM Alexa 488 and dendritic arbors were visualized with an 880 nm excitation source. For local application of 5HT to specific neuronal subcompartments or minimal 5HT application during uncaging experiments, borosilicate pipettes were filled with 200 mM

5HT in $H_2O$ (pH adjusted to 4.5). Scanning interference contrast images of slice morphology and the iontophoretic pipette were acquired with a photomultiplier tube downstream of a 770 nm longpass filter. The iotophoretic pipette was positioned upstream of the application site, relative to the overall flow of extracellular solution, either directly above the slice, near the recorded somata (within 10 µm) for uncaging experiments, or in proximity to a dendrite (0–15 µm). 5HT was applied using a 200 nA pulses (40 nA backing current) for 5 s (uncaging) or 50 ms (local iontophoresis) using an ION-100 current generator (Dagan, Minneapolis, MN).

To activate putative synaptic sites along FSI dendrites, 4-methoxy-7-nitroindolinyl-caged L-glutamate (MNI-glutamate, 2.5 mM, Tocris) was photolyzed using a 720 nm excitation source (0.5 ms duration, power determined empirically to produce 0.5–5 mV EPSPs at the soma). Five uncaging locations were chosen on a single dendrite spaced 1 µm apart. Glutamate was uncaged at each location individually and then at all locations together in a burst with varying interstimulus intervals (0.12, 5, 10, 20, 50 ms). The sequence of uncaging events always began at the most distal dendritic location, approaching the soma serially. Results were averaged over 4–7 repetitions at each interstimulus interval before and after iontophoretic 5HT application. Trials in which EPSP failures were noted due to preparation drift, or in which spikes were generated, were discarded from analysis.

## Computational modeling

### Algebraic model

A simple double exponential voltage response (template EPSP) was created using the following equation: $(1/normfac)* (-exp(-t/tau_{slow}))+exp(-t/tau_{fast}))$ where $normfac = (tau_{slow}/tau_{fast})(tau_{fast}/(tau_{fast}-tau_{slow}))$ and t is time. The decay tau ($tau_{slow}$) was set to either 15 ms ($tau_{baseline}$) or 23 ms ($tau_{5HT}$). The template EPSP was convolved five times at varying ISIs (0–50 ms). EPSP amplitude was calculated by taking the maximum of the convolved trace.

### Compartmental model in NEURON

Our compartmental neuronal model was adapted from a model FSI from the Allen Institute for Brain Science (ID#: 469610831). All channel parameters from this model were translated into NEURON and commands were run in the hoc language. The original model contained only one mechanism for passive conductance (reversal potential e = −61.6229). This single passive mechanism was split into two separate mechanisms: $K^+$ ($K_{pas}$, $e_K$ = −107 mV) and $Na^+$ passive ($Na_{pas}$, $e_{Na}$ = 53 mV) and the values of these conductances (g) were altered while maintaining their relative conductances to each other until the resting membrane potential and input resistance of the cell matched experimental values. Dendritic diameter was set to 1.5 µm (*Gulyás et al., 1999*) and axial resistance ($R_a$) was set to 172 (*Nörenberg et al., 2010*). Five double exponential synapses ($e_{syn}$ = 0 mV) were either placed at the midpoint of one dendrite (96 µm from soma, 1 µm apart). Synaptic parameters (tau, weight) were modified until the EPSP waveform matched experimental data. In some experiments, background synaptic noise was inserted as a point process into a subset of dendrites with a stochastic model containing fluctuating excitatory and inhibitory conductances (*Michalikova et al., 2017*). The level of background noise was varied by scaling total synaptic conductance to match Rin values obtained during in vivo whole-cell recordings of cortical FSIs (*Pala and Petersen, 2015*). Each model was replicated 10 times with random noise. To mimic the effects of 5HT in the model neuron, the conductances of K + channels in the soma ($K_{pas}$, Kv3.1, SK) and dendrites ($K_{pas}$, Kv3.1, Mv2) were reduced by varying percentages (90–30%) and membrane potential and input resistance (calculated by Ohm's law for a −50 pA current step) were measured. Synapses were stimulated in the model alone or at varying intervals (5, 10, 20, 50 ms). Tau of the single synaptic response was calculated by fitting an exponential to the decay of the EPSP. EPSP amplitude was calculated by taking the maximum of the voltage response.

## Immunohistochemistry

Brain slices obtained for electrophysiological recording were drop-fixed in 4% paraformaldehyde in phosphate buffered solution overnight, then rinsed with phosphate-buffered saline (PBS), cryopreserved in 30% sucrose solution in PBS, and then re-sectioned at 50 µm using a freezing stage microtome. Sections were rinsed in PBS and blocked with blocking solution (Fisher B10710) for 1 hr. Sections were then incubated in primary antibodies diluted in 0.2% TritonX-100 in PBS overnight at

4°C. The following primary antibodies were used: Rabbit anti-5HT (Immunostar 20080; 1:500) and mouse anti-GFP (Invitrogen A11120; 1:500). Sections were then rinsed with PBS and incubated in secondary antibodies (1:250, Invitrogen: Alexa 405 goat anti-rabbit A31556, Alexa 488 goat anti-mouse A11029) for 4 hr. Finally, sections were rinsed and mounted (Fisher P36934). Images were obtained using a high speed wide-field microscope (Nikon Ti, with Andor Zyla 5.5 sCMOS) with a 10x/0.45 or 20x/0.75 (+ + 60 x) Plan Apo objective. Fiji software was used to make adjustments for brightness and contrast. Widefield images were stitched with Fiji.

## Statistical analysis

All data are shown as mean ±1 SEM. We used student's t-test to compare pairs of groups if data were normally distributed (verified using Lillie test). If more than two groups were compared, we used ANOVA with post-hoc tests between groups corrected for multiple comparisons (Tukey). Sample sizes were chosen based on current standards in the field. No power analysis was done.

## Ethics statement

This study was performed in strict accordance with the recommendations in the Guide for the Care and Use of Laboratory Animals of the National Institutes of Health. All of the animals were handled according to approved institutional animal care and use committee (IACUC) protocols (AN170116, AN129822-02F) of the University of California, San Francisco. All surgery was performed under isoflurane anesthesia, and every effort was made to minimize suffering.

## Acknowledgements

This work was supported by an NRSA F31 MH111219-01 (JCA) and NIH grants U01 MH105948 (VSS) and R01 DA035913 (KJB).

## Additional information

### Funding

| Funder | Grant reference number | Author |
| --- | --- | --- |
| National Institute of Mental Health | NRSA F31 MH111219-01 | Jegath C Athilingam |
| National Institutes of Health | U01 MH105948 | Vikaas S Sohal |
| National Institutes of Health | R01 DA035913 | Kevin J Bender |

The funders had no role in study design, data collection and interpretation, or the decision to submit the work for publication.

### Author contributions

Jegath C Athilingam, Conceptualization, Data curation, Formal analysis, Funding acquisition, Validation, Investigation, Visualization, Methodology, Writing—original draft, Project administration, Writing—review and editing; Roy Ben-Shalom, Software, Formal analysis, Investigation, Methodology; Caroline M Keeshen, Investigation, Visualization; Vikaas S Sohal, Kevin J Bender, Conceptualization, Resources, Software, Supervision, Funding acquisition, Project administration, Writing—review and editing

### Author ORCIDs

Jegath C Athilingam (iD) http://orcid.org/0000-0001-5768-0105
Vikaas S Sohal (iD) http://orcid.org/0000-0002-2238-4186
Kevin J Bender (iD) http://orcid.org/0000-0001-7084-1532

### Ethics

Animal experimentation: This study was performed in strict accordance with the recommendations in the Guide for the Care and Use of laboratory Animals of the National Institutes of Health. All of

the animals were handled according to approved institutional animal care and use committee (IACUC) protocols (AN170116, AN129822-02F) of the University of California, San Francisco. All surgery was performed under isofluorane anesthesia, and every effort was made to minimize suffering.

## Decision letter and Author response

Decision letter https://doi.org/10.7554/eLife.31991.022
Author response https://doi.org/10.7554/eLife.31991.023

## Additional files

### Supplementary files
• Transparent reporting form
DOI: https://doi.org/10.7554/eLife.31991.020

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
