## [Decision Letter]

Thank you for submitting your article "Serotonin enhances excitability and gamma frequency temporal integration in prefrontal fast-spiking interneurons" for consideration by *eLife*. Your article has been favorably evaluated by Gary Westbrook (Senior Editor) and three reviewers, one of whom is a member of our Board of Reviewing Editors. The reviewers have opted to remain anonymous.

The reviewers have discussed the reviews with one another and the Reviewing Editor has drafted this decision to help you prepare a revised submission.

Summary:

The study by Athilingam et al. examines conclusively the influence of 5HT on the excitability of PV interneurons of the medial prefrontal cortex in mice using in vitro electrophysiology, optogenetics, 2P single cell imaging, pharmacogenetics and single cell computation. The authors demonstrate that 5HT has a depolarizing effect on PVIs, increases the input resistance of PV cells by reducing potassium conductances and, finally, slows the decay of EPSPs, which in turn enhances the temporal summation of EPSPs. The reviewers, however, formulated some critical points that should be addressed.

Essential revisions:

1) The effect of 5HT iontophoresis on the EPSP time course and the summation of EPSPs cannot be directly compared with the expression of hM3D receptors because rAAVs will allow an expression of hM3D receptors on the entire dendritic tree plus axon whereas the expression of 5HT receptors may be more compartmentalized. Thus, the conclusions of this set of data (hM3D receptor-mediated) should be toned down.

2) Figure 3 should include the data mentioned in the text: the change in EPSP amplitude as a function of distance along the somato-dendritic domain with and without 5HT-mediated effects (Rin, Vm).

3) The relationship between the effects of 5HT on the enhanced summation of EPSPs evoked at gamma frequencies should be experimentally strengthened. One possibility could be to examine whether the effects of 5-HT on synaptic responses translates to more rhythmic spiking output of FSIs by measuring autocorrelograms. Alternatively, they may consider inducing gamma oscillations in slice preparations (e.g. bath application of kainite) and examine whether 5HT induces enhanced gamma power. If the experimental applications are not fruitful, the authors should acknowledge the limitations of their findings.

4) Using Gq-DREADDs to examine downstream signaling pathways of 5-HT, assuming the 2A receptors are responsible for the effects of 5-HT, is an interesting approach. However, this method may not necessarily activate these signaling pathways in the same way that 5-HT does and the Gq pathway is not specific to 5-HT. The involvement of 5HT2A receptors should be directly tested. The authors should also administer CNO in uninjected mice as a control (see MacLaren et al. 2016, JNeurosci).

5) "[C]hanges in EPSP decay were most prominent in the 10-20 ms after the stimulus". The increase looks quite similar over the 10-50 ms range. Therefore, provide statistical support for this statement.

6) An outstanding question is how the effects described in the present study will translate to the intact brain. One study (Puig et al. 2010) found that 5-HT release decreased the amplitude of gamma oscillations in vivo. Please discuss more thoroughly the data in context to published work as well as future directions for examining the contribution of 5-HT to gamma oscillations.

7) Figure 1 legend seems to be wrong. Panel numbers do not line up with descriptions.

8) 50 µM 5HT seems like a high dose. Please provide some information regarding concentration-dependence of the response, and whether this is saturating.

9) The computational modeling nicely complements the experimental results. However, it seems that the conclusions regarding retention of the 5HT-enhancing effects on gamma integration in vivo need to be fleshed out. If the in vivo condition can be modeled by an increase in leak (mixed Na/K) due to synaptic iGluRs, then the relative contribution of the Kir conductance to the overall leak conductance will be less than in the in vitro case. Is this what was modeled? The same absolute change in K conductance on top of an overall increased leak?

---

## [Author Response]

Essential revisions:1) The effect of 5HT iontophoresis on the EPSP time course and the summation of EPSPs cannot be directly compared with the expression of hM3D receptors because rAAVs will allow an expression of hM3D receptors on the entire dendritic tree plus axon whereas the expression of 5HT receptors may be more compartmentalized. Thus, the conclusions of this set of data (hM3D receptor-mediated) should be toned down.

We have tempered the conclusions of the DREADD experiments by including a sentence that says, “While the Gq-DREADD activates the same downstream signaling cascades, it is not clear if it is localized to the same parts of the cell as endogenous 5HT2A receptors.” In relation to this and comment 4 below, we have performed new experiments using a specific 5HT2A receptor agonist. Results were comparable to the DREADD experiment data, suggesting that DREADD expression at the very least engages the same downstream modulatory pathways, and that off-target effects of DREADD activation are not apparent. Therefore, we feel confident that 5HT2AR signaling in FSIs enhances FSI firing in response to gamma frequency inputs, and that DREADDs are a reasonable mimic. This is important for new experiments where we utilized DREADDs to address comment 3.

2) Figure 3 should include the data mentioned in the text: the change in EPSP amplitude as a function of distance along the somato-dendritic domain with and without 5HT-mediated effects (Rin, Vm).

Since this is somewhat secondary to the main finding in Figure 3, we felt these data were best contained in a supplemental figure in the original submission (Figure 3—figure supplement 1). We prefer to leave these data as supplemental, but we welcome editorial input if you feel it should be moved to the primary figure.

3) The relationship between the effects of 5HT on the enhanced summation of EPSPs evoked at gamma frequencies should be experimentally strengthened. One possibility could be to examine whether the effects of 5-HT on synaptic responses translates to more rhythmic spiking output of FSIs by measuring auto-correlograms. Alternatively, they may consider inducing gamma oscillations in slice preparations (e.g. bath application of kainite) and examine whether 5HT induces enhanced gamma power. If the experimental applications are not fruitful, the authors should acknowledge the limitations of their findings.

To address this concern, we first examined the autocorrelograms generated from data presented in Figure 7. Unfortunately, we did not observe an increase in gamma power in these experiments, likely because of the experimental design. Here, we injected a stimulus train in which interstimulus intervals were randomly drawn from a predetermined set of ISIs. This elicited little repetitive spiking (the cell shown in Figure 7 is a good case – some cells had much less). The paucity of repetitive spiking (likely due to lack of background synaptic activity) limited our ability to observe changes in rhythmic gamma frequency spiking. Therefore, we did indeed turn to “active” slice techniques. Kainate bath application, unfortunately, did not elicit gamma at baseline. This is consistent with the literature, which has reported difficulty in generating gamma in PFC slices (though it is possible in hippocampus) (Fisahn, J Neurophys, 2004, McNally et al.,Neuroscience, 2009).

Given these results, we instead utilized an active slice preparation based on the application of carbachol, which we have successfully used in the past to elicit sustained network activity in prefrontal slices (Luongo, Horn, and Sohal, Biol Psych, 2016). Here, we recorded inhibitory responses from downstream pyramidal neurons. To isolate modulation to FSIs, we again made use of the Gq-DREADD, as this allows us to selectively activate Gq in FSIs (in contrast to 5HT2A application, which may have other effects in active slices). We found that Gq-DREADD activation of FSIs increased the probability of gamma frequency IPSCs recorded in pyramidal neurons (Figure 8). We believe this new data not only strengthens the relationship between 5HT signaling and the enhancement of gamma frequency activity in FSIs, but also provides insight into potential modulation of network activity by 5HT.

4) Using Gq-DREADDs to examine downstream signaling pathways of 5-HT, assuming the 2A receptors are responsible for the effects of 5-HT, is an interesting approach. However, this method may not necessarily activate these signaling pathways in the same way that 5-HT does and the Gq pathway is not specific to 5-HT. The involvement of 5HT2A receptors should be directly tested. The authors should also administer CNO in uninjected mice as a control (see MacLaren et al. 2016, JNeurosci).

We found that CNO has no effect when administered to brain slices from mice injected with a control fluorophore virus (AAV5-hSyn-DIO-mCherry). These data are now contained in Figure 7—figure supplement 1. As mentioned above, we have also replicated the findings of the Gq-DREADD using a 5HT2A agonist and included this in Figure 7—figure supplement 2.

5) "[C]hanges in EPSP decay were most prominent in the 10-20 ms after the stimulus". The increase looks quite similar over the 10-50 ms range. Therefore, provide statistical support for this statement.

Upon further examination, we realized that there was an error in the data analysis underlying this figure. In a few experiments, cell health diminished before the complete dataset could be acquired in 5HT. Thus, for 4 of the 43 total dendritic uncaging sites (27 total cells), there were no post-5HT data. For those conditions, baseline EPSPs were still included, but post-5HT values corresponding to the lost trials were not. For these lost trials, values of 0 were erroneously included in the original average. We have corrected this error and revised the figure. Please note that the revised figure matches the differences in mean decay far better than the original, and that there is a clear peak between 10-20 ms. Nevertheless, we cannot identify a straightforward metric to support the original statement. The wording has been changed to say “changes in EPSP decay peaked 10-20 ms after stimulation.”

6) An outstanding question is how the effects described in the present study will translate to the intact brain. One study (Puig et al. 2010) found that 5-HT release decreased the amplitude of gamma oscillations in vivo. Please discuss more thoroughly the data in context to published work as well as future directions for examining the contribution of 5-HT to gamma oscillations.

The omission of Puig et al., 2010 was an oversight and we appreciate this review. We have included information about how our work relates to this previous study and the implications for gamma oscillations in the subsection “Implications for serotonergic regulation of prefrontal circuit activity”.

7) Figure 1 legend seems to be wrong. Panel numbers do not line up with descriptions.

Thank you for catching this. It has been fixed.

8) 50 µM 5HT seems like a high dose. Please provide some information regarding concentration-dependence of the response, and whether this is saturating.

We used a dose of 30 µM in most of the manuscript (not 50 µM). We have now performed a dose-response curve experiment (Figure 1—figure supplement 2) and observed that the dose of 30 µM is sub-saturating, eliciting a response that is ~80% of maximum.

9) The computational modeling nicely complements the experimental results. However, it seems that the conclusions regarding retention of the 5HT-enhancing effects on gamma integration in vivo need to be fleshed out. If the in vivo condition can be modeled by an increase in leak (mixed Na/K) due to synaptic iGluRs, then the relative contribution of the Kir conductance to the overall leak conductance will be less than in the in vitro case. Is this what was modeled? The same absolute change in K conductance on top of an overall increased leak?

Apologies for not making this clear. We have included additional information that reads, “For these experiments, we included randomly fluctuating noise conductances that modeled both excitatory and inhibitory conductances on a subset of dendrites and stimulated synapses at baseline and after reducing K^+^ conductance by 70% as before.”